# Phosphoprotein Detection in Sweat Realized by Intercalation Structure 2D@3D g-C_3_N_4_@Fe_3_O_4_ Wearable Sensitive Motif

**DOI:** 10.3390/bios12060361

**Published:** 2022-05-24

**Authors:** Yuting Qiao, Lijuan Qiao, Peize Zhao, Peng Zhang, Fanbin Wu, Jiahui Zhang, Li Gao, Bingxin Liu, Lei Zhang

**Affiliations:** 1School of Mechanical Engineering, Qinghai University, Xining 810016, China; hiyutingqqq@126.com (Y.Q.); 18733006311@163.com (P.Z.); zhangpeng@qhu.edu.cn (P.Z.); wfb632292561@126.com (F.W.); z276475670@126.com (J.Z.); 2Research Center of Basic Medical Science, Medical College, Qinghai University, Xining 810016, China; 3Department of Mechanical Engineering, University of Alaska Fairbanks, P.O. Box 755905, Fairbanks, AK 99775-5905, USA; lzhang14@alaska.edu

**Keywords:** g-C_3_N_4_, Fe_3_O_4_, phosphoprotein, sweat sensor

## Abstract

Abnormal protein phosphorylation in sweat metabolites is closely related to cancer, cardiovascular disease, and other diseases. The real-time monitoring of phosphoproteins in sweat is significant for early monitoring of disease biomarkers. Here, a high-efficiency electrochemical sensor for phosphoprotein in sweat was realized by 2D@3D g-C_3_N_4_@Fe_3_O_4_ with intercalation structure. Common phosphoprotein β-Casein was selected to demonstrate the platform’s functionalities. The detection limit of g-C_3_N_4_@Fe_3_O_4_ could be as low as 9.7 μM, and the detection range was from 0.01 mg/mL to 1 mg/mL. In addition, the sensing platform showed good selectivity, reproducibility, and stability. We also investigated the effects of interface structure on adsorption properties and electronic properties of the g-C_3_N_4_ and Fe_3_O_4_ heterostructure using DFT. More electrons from Fe_3_O_4_ were transferred to g-C_3_N_4_, which increased the electrons in the energy band of N atoms and promoted the formation of stable N-H bonds with H atoms in phosphoproteins. We demonstrated phosphoprotein sensor functionality by measuring the phosphoprotein in human sweat during exercising. This work realizes a sensing platform for noninvasive and continuous detection of sweat phosphoproteins in wearable devices.

## 1. Introduction

Real-time noninvasive monitoring of various biomarkers in the human body has practical significance for early disease prevention and biomedical diagnosis. In daily medical testing, biological fluids, such as blood and urine, are often used for disease analysis, but it is difficult to achieve the purpose of real-time monitoring [1].

Sweat is also a kind of biological fluid, which includes common substances, such as glucose, urea, electrolytes, and lactic acid [2]. Moreover, the exocrine of human sweat contains a relatively low concentration of protein. It has been reported that the measured concentration range of proteins is about 0.3 mg/mL to 1.12 mg/mL [3], and its abnormal expression can cause diseases, such as atopic dermatitis, schizophrenia, cancer, and tuberculosis [4]. Therefore, the monitoring of protein molecules in sweat helps people to understand whether abnormal translation and expression occur.

Protein phosphorylation is one of the translational modifications in almost all cellular processes, which occurs in serine, threonine, and tyrosine. It plays an important role in physiological processes, such as cell signal transmission, cell growth cycle control, and cell apoptosis [5,6,7]. About 30% of human protein genome codes contain covalent phosphates that can be phosphorylated transiently or permanently [8,9]. Abnormal protein phosphorylation may cause Alzheimer’s disease [10], schizophrenia [11], Parkinson’s disease [12], cancer [13], cardiovascular disease [14], and other diseases. Using phosphorylated proteomics to characterize phosphorylation sites is of great significance to comprehensively understand the markers of disease diagnosis, study the underlying mechanisms, and find relevant treatment strategies.

In our work, we identified 109 different types of phosphoproteins and established their associations with different physiological states of the human body through proteomic studies using sweat as a biological sample (Appendix A). At present, many research contributions have been made to the related research on phosphorylated protein in sweat and diseases [15,16,17,18]. Therefore, it is still a relatively important opportunity to introduce a method based on the detection of phosphoproteins enrichment in sweat into the wearable electrochemical sensor.

However, it still needs to be considered that, since the protein redox centers usually exist inside three-dimensional protein shells or folded polypeptide shells, the resulting nonspecific adsorption and bioincompatibility generally lead to electrode passivation [19,20]. Therefore, the construction of protein electrochemical biosensors still has certain challenges.

In recent years, semiconductor nanomaterials represented by graphitic carbon nitride (g-C_3_N_4_) have attracted extensive attention due to their excellent chemical stability, thermal stability, catalytic activity, photoelectric properties, and biocompatibility [21]. The g-C_3_N_4_ is an analog of graphite, mainly composed of p-conjugated graphite [22], in which the covalent atomic layers between carbon and nitrogen are connected to each other by van der Waals forces [23]. However, pure g-C_3_N_4_ has poor conductivity and fewer active sites. Thus, the electronic structure of pure g-C_3_N_4_ framework can be changed by doping with heteroatoms, such as nitrogen, and promote electrochemical performance [24].

The exocrine secretions in human sweat have very low levels of protein and even lower of phosphoproteins. Therefore, a material with specific adsorption capacity for phosphoproteins is required to achieve better enrichment of phosphoproteins in sweat. To date, the technology of functionally modifying magnetic nanomaterials has been verified to be used to efficiently enrich and analyze low-abundance phosphoproteins [25]. Among them, Fe_3_O_4_ nanoparticles, as one of the most important nanostructure materials, can be highly sensitive and selective when combined with phosphate groups, mainly relying on the interaction between the Lewis acid and base [26].

Here, we report a 2D@3D g-C_3_N_4_@Fe_3_O_4_ composite with an intercalation structure for electrochemical detection of phosphoprotein in sweat. The synthesized procedure for the g-C_3_N_4_@Fe_3_O_4_ sensor elements is illustrated in Figure 1. In addition, β-Casein was used as a model protein to study the coordination mechanism of the structure-activity relationship between phosphoproteins and g-C_3_N_4_@Fe_3_O_4_. A facile electrochemical strategy for real-time monitoring of phosphoproteins in sweat was developed, which could be further broadly applied in the healthcare field.

## 2. Results and Discussion

### 2.1. Characterization of the g-C_3_N_4_@Fe_3_O_4_ Composite

In this study, a sweat sensor was constructed with g-C_3_N_4_@Fe_3_O_4_ intercalation structure materials (Figure 1) as sensitive primitives, with a subsequent application in the real-time monitoring of phosphoprotein. A nitrogen-doped g-C_3_N_4_ was synthesized using melamine as raw material and urea as nitrogen source. It could effectively fill the nitrogen vacancy defect structure of g-C_3_N_4_ formed by melamine and further improve the conductivity of g-C_3_N_4_. Appendix A presents the morphology of melamine, which exhibited predominantly random particle shapes. Additionally, Appendix A presents a heterogeneous layered structure of urea. The g-C_3_N_4_ intermediate formed through the hydrothermal reaction of melamine and urea is shown in Appendix A. Among them, melamine and urea further formed nanopillar clusters through the van der Waals stacking reaction. With the occurrence of hydrothermal reaction, the melamine molecule and the urea molecule underwent a self-assembly reaction, and the melamine was promoted to transform into a nano-columnar-like structure with a diameter of about 1.3 μm. In addition, the interior of the nanocolumnars was continuously stacked by hydrogen bonds, which made the intermediates gradually pile up to form the aligned nanopillar-like cluster structure.

Figure 1a is a SEM image of the nitrogen-doped g-C_3_N_4_ nanomaterials prepared by calcining the intermediate to form an intercalation structure. Interestingly, it can also be clearly observed from Figure 1b that g-C_3_N_4_ showed sheet/tube/sheet structures. This is because the hydrogen bonds between the inner layers of the nanocolumnar cluster intermediates are destroyed by the gradual increase in temperature during the heating process of calcination. This further leads to transverse stripping of the intermediates to form isolated sheet/tube/sheet intercalated structures. In addition, these nanotube clusters were uniformly arranged on a coplanar surface, which could effectively improve the stability of the g-C_3_N_4_ structure and facilitate the fast electron transport between nanotubes. The structure of g-C_3_N_4_ nanomaterials was consistent with TEM observation. As shown in Figure 1c and Appendix A, with the prolongation of calcination time, the interior of the intermediate exfoliation layer gradually transformed into a nanotube cluster structure with a diameter of about 25 nm through self-rolling. Moreover, many curved nanotube clusters are arranged uniformly and orderly among the nanosheets. The preparation process of g-C_3_N_4_ obviously underwent multi-step structural changes, in which the transverse stripping of the intermediates was the fundamental reason for reducing the diameter of g-C_3_N_4_ tubes [27].

Then, the Fe_3_O_4_ nanoparticles were in situ embedded on g-C_3_N_4_ surfaces (Figure 1d) by co-precipitation with FeSO_4_·7H_2_O and FeCl_3_·6H_2_O, which have a specific adsorption capacity for phosphoproteins. The original intercalation structure of g-C_3_N_4_ was still maintained. The reason considered was that after ultrasonic treatment, the weak bonds inside the g-C_3_N_4_ were destroyed, so that many nanowire structures were formed on the basis of the original structure. The composite materials restrained each other while maintaining a uniform dispersion state, forming a heterojunction structure and effectively solving the agglomeration problem of nanoparticles. Overall, a point-line-plane trinity intercalation structure composite material structure was formed.

X-ray diffraction patterns in wide angle of the g-C_3_N_4_@Fe_3_O_4_ composite (Figure 2a) displayed typical diffraction peaks of g-C_3_N_4_ and Fe_3_O_4_, which could be indexed to JCPDS PDF#87-1526 and JCPDS PDF#99-0073. Specifically, the peaks at 12.7° and 27.5° were assignable to the diffractions of triazine unit structure stacking (100) and crystal and aromatic ring stacking (002) interlayer planes of g-C_3_N_4_ [28,29]. Moreover, the powder diffraction data of Fe_3_O_4_ particles include the characteristic peaks at 30.1°, 35.4°, 43.1°, 53.4°, 56.9°, 62.5°, and, respectively, belonged to (220), (311), (400), (422), (511), (440) planes [30]. This showed that the introduction of different ratio of Fe_3_O_4_ inhibits the superposition of g-C_3_N_4_ in the vertical direction of the crystal plane, and there is no obvious peak shift.

FT-IR was performed as shown in Figure 2b. The absorption peak of g-C_3_N_4_ at 3186 cm^−1^ is caused by the N-H stretching vibration peak and the O-H vibration mode, where the surface absorbs moisture. The multiple absorption peaks located at 1241 cm^−1^, 1320 cm^−1^, 1409 cm^−1^, 1458 cm^−1^, 1567 cm^−1^, 1635 cm^−1^ correspond to the characteristic peaks of the aromatic C-N heterocyclic ring and stretching vibration. Additionally, the absorption peak at 823 cm^−1^ belongs to the stretching vibration mode of the triazine ring structure molecule [31,32,33]. The absorption peak with a wavenumber of 570 cm^−1^ is caused by the Fe-O vibration of pure Fe_3_O_4_ [34]. Above all, the FT-IR spectra indicated the successful formation of the g-C_3_N_4_@Fe_3_O_4_ composites.

The thermogravimetric analysis in Figure 2c explored the thermal stability of the material in an air atmosphere (Appendix A). The results showed that the thermal stability of the sensing materials could meet the standard of application at room temperature. The hysteresis regression line performance at room temperature is shown in Figure 2d. The results showed that all nanomaterials are generally superparamagnetic (Appendix A).

XPS spectra of CNFeO-0.4 was performed as shown in Figure 3a. The peaks at 725.85 eV and 711.85 eV correspond to Fe2p_1/2_ and Fe2p_3/2_ of Fe_3_O_4_, respectively. Additionally, the peaks at 399.85 eV and 284.85 eV correspond to N1s and C1s of g-C_3_N_4_, respectively. Figure 3b shows the binding energy peaks at 284.8 eV, 286 eV, and 288.7 eV, corresponding to the C1s, which were, respectively, expressed as the surface-activated carbon C-C structure, C-N-C position bond structure, and sp^2^ hybrid carbon C-(N)_3_ structure [35,36,37]. Figure 3c shows the binding energy peaks at 399.1 eV, 400.5 eV, and 404.8 eV, corresponding to the N1s, which were, respectively, expressed as the tertiary amine pyrrole nitrogen N-(C)_3_ structure, C-N-H structure, and π-excitation [38,39,40]. Figure 3d shows that the Fe2p peak could be disassembled into two strong characteristic peaks of Fe2p_3/2_, Fe2p_1/2_ and two satellite peaks. The two photoelectron peaks of 711.65 eV and 724.9 eV are attributed to Fe^3+^, whereas 717.5 eV and 732.6 eV, respectively, correspond to two satellite peaks, further indicating the existence of Fe_3_O_4_ [41,42]. Figure 3e shows the binding energy peaks at 529.5 eV, 530.5 eV, and 531.5 eV, corresponding to the O1s. It came from two different compounds in the sample linked to F-O, C-OH, and C-O-C structures [43,44,45]. These results prove the coexistence of g-C_3_N_4_ and Fe_3_O_4_ in the composite.

Figure 3f is the nitrogen adsorption-desorption isotherms of g-C_3_N_4_, Fe_3_O_4_, and g-C_3_N_4_@Fe_3_O_4_. The results show that the synthesized composite materials have a large specific surface area of 85.895 m²/g, which was significantly higher than that of g-C_3_N_4_ (53.338 m²/g). It is caused by the presence of Fe_3_O_4_ particles, which makes the exfoliation degree of the intercalation structure of g-C_3_N_4_ more dispersed. Thus, the doping of Fe_3_O_4_ could provide more active sites of g-C_3_N_4_, which was beneficial for improving the electron transport ability (Appendix A). The results of Zeta potential show that the isoelectric point of the g-C_3_N_4_@Fe_3_O_4_ was pH = 5.747. (Appendix A).

### 2.2. Phosphoprotein Sensing

The electron transfer ability of different g-C_3_N_4_@Fe_3_O_4_ was analyzed by electrochemical impedance spectroscopy (EIS). As shown in Figure 4a, the CNFeO-0.4 possessed the best electron transfer ability. Therefore, CNFeO-0.4 was selected as the sensing material in the subsequent experiments. It can be seen from Appendix A that the CV response of pure g-C_3_N_4_ is significantly lower than that of g-C_3_N_4_ doped with urea as the nitrogen source. Likewise, the EIS electron transfer ability of nitrogen-doped g-C_3_N_4_, as shown in Appendix A, is significantly higher than that of pure g-C_3_N_4_.

A pair of obvious redox peaks can be found in curve I (bare GCE) on the bare electrode, and the peak potential difference is about 110.0 V. The gold nanoparticles were modified on the surface of the bare electrode by electrochemical potentiation deposition, and the current intensity of curve II (Au/GCE) increased significantly to the largest current density. The reason is that the gold nanoparticles deposited by potentiostatic have excellent electrical conductivity, which promotes the rapid transfer of electrons [46,47]. g-C_3_N_4_ and Fe_3_O_4_ are relatively unfavorable for the transfer of electric charges, leading to a decrease in the current values of curves IV (g-C_3_N_4_/Au/GCE) and V (Fe_3_O_4_ /Au/GCE), respectively. The negative charge of g-C_3_N_4_ would form a repulsion effect with cyanide ions, and the wide band gap (2.7 eV) of g-C_3_N_4_ restricts and hinders electron transfer [48]. The relatively poor conductivity of Fe_3_O_4_ leads to increased resistance to electron transfer in the redox process [49]. The current of curve III (g-C_3_N_4_@Fe_3_O_4_) when forming a heterojunction was significantly higher than that of curves IV and V, respectively. This is because the g-C_3_N_4_@Fe_3_O_4_ heterojunction can promote the transfer ability of electrons [50]. A strong Au-S covalent bond was formed on the active sites of gold on the electrode surface [51] after the dropwise addition of the MCH blocking solution (6-mercapto-1-hexanol). Thus, it could be observed that the current intensity of curve VI (MCH/g-C_3_N_4_@Fe_3_O_4_/Au/GCE), VII (MCH/g-C_3_N_4_/Au/GCE), and VIII (MCH/ Fe_3_O_4_/Au/GCE) was significantly lower than that of curves III, IV, and V. Comparing the current intensity of curve IX (β-Casein/MCH/g-C_3_N_4_@Fe_3_O_4_/Au/GCE), X (β-Casein/MCH/g-C_3_N_4_/Au/GCE), and XI (β-Casein/MCH/Fe_3_O_4_/Au/GCE), it was found that the current intensity is further reduced. This is because β-Casein is a macromolecular protein, which hinders the effective area and active site of electron transfer in electrochemistry after specific adsorption [52]. In summary, it could be found that each step of the electrode modification process was relatively successful, and β-Casein was successfully adsorbed on the electrode surface.

Figure 4c and Appendix A are the EIS of electrode under different modification conditions. The charge transfer resistance (Rct) of curve I was about 118.0 Ω. The Rct of curve II was significantly reduced to approximately 14.5 Ω, which revealed that the gold nanoparticles promote the conductivity of the electrode. The Rct was significantly increased after the electrode was modified with g-C_3_N_4_@Fe_3_O_4_, g-C_3_N_4_, and Fe_3_O_4_, which were approximately 105.2 Ω, 417.7 Ω, and 532.2 Ω for curve III, IV, and V, respectively. The EIS of the constructed electrode further increased, since the Rct of curve VI, VII, and VIII were about 577.0 Ω, 764.57 Ω, and 992.4 Ω after the MCH was added to the seal. The Rct of curve IX, X, and XI were about 1463.1 Ω, 2055.9 Ω, and 2504.2 Ω, respectively. This was because the β-Casein hindered the transfer of electrons to a greater extent after the specific adsorption on the electrode surface.

The electrochemical reaction kinetics of β-Casein was examined on β-Casein/MCH/g-C_3_N_4_@Fe_3_O_4_/Au/GCE via CV with different scoped scan rates. In Figure 4d, both the oxidation peak currents (Ipa) and reduction peak currents (Ipc) increased with increasing scan rate from 10 to 200 mV/s. This phenomenon revealed the typical quasi-reversible electron transfer dynamics on the electrode [53]. The relationship between the response of the peak current change and the square root of the scan rate is shown in Appendix A. The linear fit equations were expressed as I_pa_ = 8.69528X + 2.91385 (R^2^ = 0.99961) and I_pc_ = −5.91195X − 10.11222 (R^2^ = 0.99975), respectively. The fitting result means that the redox reaction of β-Casein at β-Casein/MCH/g-C_3_N_4_@Fe_3_O_4_/Au/GCE was an adsorption control process [54].

Figure 4e shows the CV response of MCH/g-C_3_N_4_@Fe_3_O_4_/Au/GCE with different concentrations of β-Casein. As the concentration of β-Casein increased, the oxidation peak and reduction peak currents were significantly reduced. The concentration increased from 0.1 mg/mL to 1 mg/mL, and the oxidation peak and reduction peak currents changed from 121.8 μA to 34.85 μA, and −100.9 μA to −27.53 μA, respectively. In Figure 4f, the linear fit equations were I_pa_ = −90.92121X + 130.27267 (R^2^ = 0.98551) and I_pc_ = 71.85455X − 103.062 (R^2^ = 0.96568), respectively. The R^2^ of these fitting equations was close to 1, indicating that the obtained formulae were more accurate.

In Appendix A, the correlation between different concentrations of β-Casein and the electrode current response was further determined by differential pulse voltammetry (DPV). It can be seen that, as the concentration of β-Casein increased from 0.1 mg/mL to 1 mg/mL, the peak current also decreased from 61.08 μA to 18.87 μA, which was consistent with the overall performance of the CV. The linear fit equation obtained in Appendix A was Y = −44.5624X + 64.59933 (R^2^ = 0.99221). The R^2^ of the fitting equation was close to 1, indicating that the obtained formulae were more accurate. The limit of detection (LOD) was calculated by referring to Equation (1).
(1)$$\sigma_{\mathrm{LOD}}=\frac{3\mathrm{SD}}{m}$$
where SD is the standard deviation value of the blank sample, and m is the slope value of the standard curve of the test substance. According to the calculation, the detection limit of the sensor is 9.74 μM.

The electrochemical sensing performance of the β-Casein/MCH/g-C_3_N_4_@Fe_3_O_4_/Au/GCE was further compared with previously reported β-Casein sensors (Appendix A). As shown, the β-Casein/MCH/g-C_3_N_4_@Fe_3_O_4_/Au/GCE had lower LOD. Furthermore, it is worth mentioning here that our work is the first to achieve the detection of phosphoproteins in sweat over a range of protein concentrations in sweat.

### 2.3. The Sensing Mechanism Analysis

From the above discussion on the structure and properties of g-C_3_N_4_@Fe_3_O_4_, in order to understand the interaction of the constructed composites and β-Casein, the density functional theory (DFT) was used to study the synergistic mechanism (Figure 5).

Figure 5a represents the interaction of nitrogen-doped g-C_3_N_4_ (gray-purple structural chain) and β-Casein, where red, crimson, brown, yellow, and purple balls represent O atom, H atom, C atom, S atom, and N atom. As shown in Figure 5a, a large number of H atoms in β-Casein closely adhere to N atoms in g-C_3_N_4_ to form hydrogen bonds (N-H bonds). Here, the nitrogen-rich graphitic g-C_3_N_4_ synthesized with urea doped with melamine possessed more N atoms on the g-C_3_N_4_ surface, which effectively filled up many nitrogen vacancies formed on the surface of g-C_3_N_4_ synthesized from pure melamine. The interaction between Fe_3_O_4_ (gold-red chain structure) and β-Casein is shown in Figure 5b, in which the red and gold balls represent the O and Fe atoms, respectively. In Figure 5b, it is found that only the O atoms in β-Casein are adsorbed to the Fe atoms in Fe_3_O_4_. The Fe atoms would capture a large number of electrons from the O atoms to form Fe-O bonds, resulting in an increase in electronegativity of the O atoms, and the empty orbital could not form a hydrogen bond with the H atoms of the β-Casein. Therefore, the interaction effect is relatively weaker than that of g-C_3_N_4_. It can be found that the interaction between g-C_3_N_4_ and β-Casein is better.

Figure 5c shows the interaction of g-C_3_N_4_@Fe_3_O_4_ with β-Casein. The interaction was mainly through functional groups, such as N-C=N, N-(C)_3_, and C-N-H in g-C_3_N_4_@Fe_3_O_4_ and the amino group (-NH_2_) of free amino acids in the shell of β-Casein protein to form N-H bonds. In addition, the larger specific surface area of the g-C_3_N_4_@Fe_3_O_4_ composite also provided more active sites for the action of β-Casein. On the other hand, g-C_3_N_4_ and Fe_3_O_4_ could form a p-n-type heterostructure at the composite interface, which made the carriers have better spatial separation. It was conducive to the rapid transfer of electrons, and the overall synergy was presented. In addition, the electron transfer between g-C_3_N_4_@Fe_3_O_4_ composite was through the Fe-N bonds. Structurally, there were many C-N bonds, C-C bonds, C-H bonds, C-O bonds, O-H bonds, N-H bonds, and a small amount of C-S bonds in the β-Casein. In addition, there was a large number of C-N bonds and Fe-O bonds in g-C_3_N_4_ and Fe_3_O_4_, respectively. The bond lengths of g-C_3_N_4_, Fe_3_O_4_ and g-C_3_N_4_@Fe_3_O_4_ interacting with β-Casein are shown in Appendix A, respectively. Additionally, the adsorption energy changes are shown in Appendix A. As shown in Appendix A, the bond length of g-C_3_N_4_@Fe_3_O_4_ increased significantly after the interaction with β-Casein, which can be seen by comparing the bond length after the interaction of g-C_3_N_4_. In addition, the C-N bonds, Fe-O bonds in g-C_3_N_4_ and Fe_3_O_4_, as well as the N-H bond lengths formed by g-C_3_N_4_ and β-Casein were significantly increased (Appendix A). This shows that the interaction between the composite and β-Casein was stronger, and the β-Casein was activated to a higher degree. It can also be seen from Figure 5c that g-C_3_N_4_@Fe_3_O_4_ interacted with β-Casein, and the atoms in the g-C_3_N_4_ structural chain were more closely arranged. More H atoms in the structure moved toward N atoms to form many N-H bonds. However, Fe_3_O_4_ would transfer more electrons to g-C_3_N_4_ through the Fe-N bonds formed in g-C_3_N_4_@Fe_3_O_4_, and the number of electrons in the N atomic energy band was higher. The result was that the N-H bonds formed with the H element in β-Casein were more stable, which enhanced the interaction effect of β-Casein to a greater extent.

In addition, the adsorption energies of all materials with β-Casein were negative, indicating that the adsorption process was exothermic. Therefore, the adsorption process was stable (Appendix A).

### 2.4. Selectivity, Reproducibility, and Stability

#### 2.4.1. Selectivity

The selectivity of the sensor toward β-Casein was evaluated in the presence of various common nonphosphoproteins (bovine serum albumin, α-lactalbumin, and β-lactoglobulin) with the ratio of 1:100 (V_β-Casein:_V_mixed nonphosphoproteins_). Figure 6a,b indicate that the g-C_3_N_4_@Fe_3_O_4_ showed a significant response to β-Casein mixed with nonphosphoproteins but an inconspicuous response to these individual nonphosphoproteins. The results showed that the constructed phosphoprotein electrochemical sensor could selectively respond to phosphoprotein even under the complex system with different concentrations of nonphosphoprotein interference.

#### 2.4.2. Reproducibility

The reproducibility of g-C_3_N_4_@Fe_3_O_4_ was performed in an electrolyte solution five times. The CV and current peak histogram results are shown in Figure 6c,d. The electrode-detected peak potentials were basically the same. The calculated relative standard deviation (RSD) of the measured results was 2.17%, which indicates that the constructed phosphoprotein electrochemical sensor has good reproducibility.

#### 2.4.3. Stability

The stability of the constructed sensor was tested by the same electrode with 10 mg/L β-Casein five consecutive times (Figure 6e,f). The RSD of the calculated peak current was 0.59%, which shows that the constructed phosphoprotein electrochemical sensor has good repeatability.

### 2.5. Practical Application Sensor Evaluation

In order to verify the accuracy of this method, sweat monitoring in practical applications was performed by attaching a chip (Figure 7a). The specific device construction process is based on a PI flexible substrate chip with Au as the working electrode, Ag/AgCl as the reference electrode, and Pt as the counter electrode. It was further completed by spinning g-C_3_N_4_@Fe_3_O_4_ sensing material and coating MCH blocking solution to block unreacted active sites and Tween-20 active agent solution. By constructing a g-C_3_N_4_@Fe_3_O_4_ sensing array on a PI flexible three-electrode, the electrochemical chip was attached to the volunteer’s curved arm without any separation (Figure 7b). An application of the CV response of the sensor was obtained when a person cycled in a closed room for 15 min while drinking 100 mL milk, while the contrastive CV response was obtained from another person cycling for 15 min. The CV response of the former was significantly lower than that of the latter, as shown in Figure 7c. This was attributed to the macromolecular phosphoprotein metabolized from sweat adsorbed on the surface of g-C_3_N_4_@Fe_3_O_4_. These results indicate that the constructed phosphoprotein electrochemical sensor was potentially applicable for phosphoprotein in sweat in the real sample.

## 3. Conclusions

In conclusion, we developed a g-C_3_N_4_@Fe_3_O_4_ intercalated composite as a specificity-sensitive moiety by hydrothermal synthesis, calcination, and co-precipitation. By constructing a sensing array on a flexible sensing platform, we demonstrated a skin-adhering, noninvasive, real-time, and in situ wearable sensing platform for sweat phosphoprotein monitoring. The detection limit of this electrochemical sensing platform could be as low as 9.7 μM, and the detection range was 10 mg/L to 1mg/mL. By combining the results of the adsorption between the intercalation structure g-C_3_N_4_@Fe_3_O_4_ composite and the β-Casein with the density functional theory calculation, a better scientific rational explanation for the detection mechanism was provided. Thus, our sweat phosphoprotein monitoring wearable platform further enables applications in clinical monitoring and precision medicine. Combining spatial network big data and artificial intelligence technology, such as high-precision medical biosensors, could be further studied in real-time, noninvasive sweat monitoring and sweat metabolism phosphoproteomics.

## Data Availability

Not applicable.

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
