# Peer review of "Phosphoprotein Detection in Sweat Realized by Intercalation Structure 2D@3D g-C3N4@Fe3O4 Wearable Sensitive Motif"

_biosensors, 2022, doi:10.3390/bios12060361_

Round 1
Reviewer 1 Report
In this work, the authors successfully developed a high-efficiency electrochemical sensor for phosphoprotein in sweat was realized by 2D@3D g-C3N4@Fe3O4 with intercalation structure. In addition, β-Casein was used as a model protein to study the coordination mechanism of structure-activity relationship between phosphoproteins and g-C3N4@Fe3O4. I would like to recommend it for publication after appropriate revisions. The detailed comments are as follows:
1. The author emphasized the sample CNFeO-0.4 without describing the influence of the different ratio of the introduction of Fe3O4. It is suggested to add some comparison of different samples.
2. In the analysis of XRD part, why did the peak at 12.7o of g-C3N4 disappear after the introduction of Fe3O4? Is there any explanation of this phenomenon?
3. In the XPS analysis part, the author mentioned that “The absorption peak with a wavenumber of 570 cm-1 is caused by the Fe-O vibration of pure Fe3O4 that the Fe3O4 is successfully embedded on the g-C3N4 with increasing ratio of iron source.” However, the typical peak caused by the Fe-O vibration can only be vaguely observed in sample CNFeO-0.2, 0.3, 0.4, I don’t think the description is right.
4. In order to prove the feasibility, it is suggested that the author should make a series of comparative analysis between this work and other similar studies.
5. The important papers should be cited in this paper. ”Angew. Chem. Int. Ed., 2021, 60, 25318‘’.
Author Response
Respond to the Reviewers
Reviewer #1
1. The author emphasized the sample CNFeO-0.4 without describing the influence of the different ratio of the introduction of Fe3O4. It is suggested to add some comparison of different samples.
Response: Thanks for your suggestion. It should be noted here that the intercalation structure composite studied in this experiment is co-precipitated and compounded with intercalation structure g-C3N4 in different proportions under the condition that the amount of iron salt remains unchanged.
Actually, we have described the preparation of g-C3N4@Fe3O4 with different ratios in the supporting information (1. EXPERIMENTAL PROCEDURES, 1.9 Preparation of g-C3N4@Fe3O4). Furthermore, in the manuscript we have explained the FT-IR and XRD structural characterizations of g-C3N4@Fe3O4 with different ratios. Moreover, we performed electrochemical impedance spectroscopy tests on g-C3N4@Fe3O4 materials with different ratios, and the results showed that CNFeO-0.4 exhibited the best electron transfer properties. Therefore, the ratio of CNFeO-0.4 was determined as the sensitive material in subsequent experiments.
2.In the analysis of XRD part, why did the peak at 12.7oof g-C3N4 disappear after the introduction of Fe3O4? Is there any explanation of this phenomenon?
Response: Thanks for your suggestion. By normalizing the XRD pattern, we can clearly see that the peak at 12.7o belonging to g-C3N4 still exists.
Similarly, it should be pointed out that the XRD peaks of g-C3N4 and Fe3O4 did not shift significantly on the whole. With the increase of the input amount of g-C3N4, the characteristic peaks of g-C3N4 increased, and the characteristic peaks of Fe3O4 changed slightly and gradually decreased. In addition, we have redrawn the XRD patterns in the manuscript and in the reply (Figure. 2a).
3.In the XPS analysis part, the author mentioned that “The absorption peak with a wavenumber of 570 cm-1is caused by the Fe-O vibration of pure Fe3O4 that the Fe3O4 is successfully embedded on the g-C3N4 with increasing ratio of iron source.” However, the typical peak caused by the Fe-O vibration can only be vaguely observed in sample CNFeO-0.2, 0.3, 0.4, I don’t think the description is right.
Response: Thanks for your suggestion. We reprocessed the FT-IR spectra by normalization. It can be clearly seen in the Figure (orange stripes) that the absorption peaks caused by Fe-O are obvious at the wavelength of 570 cm-1 after normalization.
In addition, we also made corrections to the description in the original text [1]-[3]. “The absorption peak with a wavenumber of 570 cm-1 is caused by the Fe-O vibration of pure Fe3O4 that the Fe3O4 is successfully embedded on the g-C3N4 with increasing ratio of iron source.” has been revised as“Above all, the FT-IR spectra indicated the successful formation of the g-C3N4@Fe3O4 composites.”
[1] Qi, X.; Liang, L.; Jia, W.; Guo, X.; Jing, D.; Dong, F; Jian, L. The magnetically separable Pd/C3N4/Fe3O4 nanocomposite as a bifunctional photocatalyst for tetracycline degradation and hydrogen evolution. Colloids and Surfaces A: Physicochemical and Engineering Aspects 2022, 641, 128404.
[2] Man, W.; Hao. Y.; Wen. D.; Wen. B.; Xiao. Y.; A Taiji-principle-designed magnetic porous C-doped graphitic carbon nitride for environment-friendly solid phase extraction of pollutants from water samples. Journal of Chromatography A 2015, 1412, 12-21.
[3] Y. W.; M. C.; X. Y.; J. R.; Y. D.; J. W.; J. P.; Y. W.; X. C. Hydrothermal synthesis of Fe3O4 nanorods/graphitic C3N4 composite with enhanced supercapacitive performance. Materials Letters 2017, 198, 114-117.
4. In order to prove the feasibility, it is suggested that the author should make a series of comparative analysis between this work and other similar studies.
Response: Thanks for your suggestion. Based on your comments, we have performed a series of comparative analyses of this work with similar studies in other fields (Table. S3) and explained in the manuscript. It is worth mentioning here that our work is the first to achieve detection of phosphoproteins in sweat over a range of protein concentrations in sweat.
Table. S3 Comparison of the electrochemical sensing performance of different field toward phosphoprotein detection.
|
Detection System |
Sensor Materials |
LOD |
Liner Range |
References |
|
Water |
ZnII-DPAa |
0.22 ppm |
|
[1] |
|
Glioblastoma cell |
Silicon photonic microring resonator arrays |
0.6 pM |
3.55-log |
[2] |
|
Food |
NH2-TiO2/UCNPsb-rGO |
9.2 × 10−5 mg/mL |
0-1 mg/mL |
[3] |
|
Electrolyte |
DPA-Zn2+c |
|
≥1 nM |
[4] |
|
Food
|
NH2-TiO2/MUAd/AuEe-QCMf |
0.09 mM |
1.0×10-3 -1.0 mg /mL |
[5] |
|
Cancer cell |
Zr-FeTCPPg-MOF |
|
0.1–40 nM/40–150 nM |
[6] |
|
Electrolyte |
DPA-NH2h |
|
≥1 nM |
[7] |
|
Sweat |
g-C3N4@Fe3O4 |
9.7 μM |
0.01-1mg/mL |
This work |
a: Zinc(II)-dipicolylamine; b: Upconversion nanomaterials; c: Dipicolylamine–zinc chelates 4; d: 11-mercaptoundecanoic acid; e: Au electrode; f: Quartz crystal microbalance 5; g: Fe (III) meso-Tetra (4-carboxyphenyl) porphine 6; h: 4-[bis(2-pyridylmethyl)aminomethyl]aniline
[1] Tsukuru, M.; Tsuyoshi, M.; Petr, K.; Pavel, Jr.; Shi, T. Antibody- and Label-Free Phosphoprotein Sensor Device Based on an Organic Transistor. Anal. Chem. 2016, 88, 1092–1095.
[2] James, W.; Aurora, A.;, Nicholas, V.; Hongi, Y.; Mark, J.; Ryan, B. Rapid, Multiplexed Phosphoprotein Profiling Using Silicon Photonic Sensor Arrays. ACS Central Science 2015, 1, 374−382.
[3] Jian, G.; Shi, L.; Shuo, W.; Jun, W. Determination of Trace Phosphoprotein in Food Based on Fluorescent Probe-Triggered Target-Induced Quench by Electrochemiluminescence. Journal of Agricultural and Food Chemistry 2020, 68, 12738-12748.
[4] Saima, N,; Mubarak, Al.; Ishtiaq, A.; Christof, N.; Wolfgang, E. Biomolecular Detection with a Single Nanofluidic Diode Decorated with Metal Chelates. ChemPlusChem 2020, 85, 101002.
[5] Jian, G.; Guo, F.; Shuo, W.; Jun, W. Quartz crystal microbalance sensor based on 11-mercaptoundecanoic acid self-assembly and amidated nano-titanium film for selective and ultrafast detection of phosphoproteins in food. Food Chemistry 2021, 344, 128656.
[6] Xin, L.; Shuang, E.; Xu, C. Metal-organic framework/3,30,5,50-tetramethylbenzidine based multidimensional spectral array platform for sensitive discrimination of protein phosphorylation. Journal of Colloid and Interface Science 2021, 602 , 513–519.
[7] Saima, N.; Mubarak, A.; Ishtiaq, A.; Christof, N.; Wolfgang, E.; Phosphoprotein Detection with a Single Nanofluidic Diode Decorated with Zinc Chelates. ChemPlusChem 2020, 85, 587–594.
5. The important papers should be cited in this paper. “Angew. Chem. Int. Ed., 2021, 60, 25318”.
Response: Thanks for your suggestion. We have cited this paper as reference [53].

Reviewer 2 Report
In this manuscript, hybrid g-C3N4@Fe3O4 structures were prepared for binding with beta-Casein. Wearable electronic was constructed for realizing the sweat-sensing application.
Please discuss why the reported structure can be bound specifically to beta-Casein but not to other proteins?
In Figure 2, the XRD patterns for all structures with Fe3O4 are unclear. The signals are too weak. These curves must be improved.
In scheme 1, the graphics are very difficult to follow. For example, why melamine is a lump; intermediate is rod-shaped; FeSO4 has five bonds? What is the molecule consisted of 3 purple balls and 6 grey balls? This synthetic scheme must be coherent to the synthetic procedures reported.
In scheme 2, what are C6H14OS and C26H50O10? The fabrication of the wearable structure must be stated and discussed. Figure S8 and Figure S9 (curves only) should be moved to the main text.
More references on iron oxide-gold hybrid structures should also be cited.
The English usage must be improved. There are many typos in the manuscript, for example:
Scheme 2, caption: diagaram; sewat
Author Response
Respond to the Reviewers
Reviewer #2
1. Please discuss why the reported structure can be bound specifically to beta-Casein but not to other proteins?
Response: Thanks for your suggestion. In previous sweat metabolomics studies (Supporting Information Table. S2), sweat contained more casein-based phosphoproteins. For example, Protein kinase C and casein kinase substrate in neurons protein 3, Protein kinase C and casein kinase substrate in neurons protein 2, and Serine protease HTRA3. These proteins could be involved in endocytosis, regulate the internalization of plasma membrane proteins and act as tumor cell suppressors.
β-Casein, as one kind of basic phosphoprotein, can be effectively absorbed on metal oxides. Metal oxides represented by Fe3O4 have good physical and chemical stability, so Metal Oxide Affinity Chromatography (MOAC) method has gradually become the main method for enriching phosphopeptides. Studies have shown that the surfaces of metal oxides have amphoteric properties due to the valence unsaturation of metal and oxygen atoms. That is to say, at low pH value, the Lewis acid is positively charged and can effectively combine with negatively charged phosphate groups, which is the adsorption of phosphoproteins on metal oxides. At high pH, the Lewis basicity will elute the phosphoprotein adsorbed on the metal oxide. Due to the amphiphilic nature of the surface, these metal oxides have strong reversible binding force to phosphate, which can be used for the enrichment of phosphorylated peptides[1]-[5].
In addition, graphitic carbon nitride (g-C3N4) is the most stable allotrope of carbon nitride. Related research shows that g-C3N4 nanomaterials can bind with phosphate groups[6].
[1] Fei, W.; Ya, G.; Sen, Z.; Yan, X. Hydrophilic modification of silica–titania mesoporous materials as restricted-access matrix adsorbents for enrichment of phosphopeptides. Journal of Chromatography A 2012, 1246 , 76–83.
[2] Wei, C.; Yu, C. Functional Fe3O4@ZnO magnetic nanoparticle-assisted enrichment and enzymatic digestion of phosphoproteins from saliva. Anal Bioanal Chem 2010, 398, 2049–2057.
[3] Dong, X.; Guo, Y.; Ming, G.; Chun, D.; Xiang, Z.. Selective Enrichment of Glycopeptides/Phosphopeptides Using Fe3O4@Au- B(OH)2@mTiO2 Core-Shell Microspheres. Talanta 2017, 166, 154-161.
[4] Jin, L.; Meng, W.; Chun, D.; Xiang, Z. Facile synthesis of Fe3O4@mesoporous TiO2 microspheres for selective enrichment of phosphopeptides for phosphoproteomics analysis. Talanta 2013, 105, 20–27.
[5] Ying, Y.; Jin, L.; Chun, D.; Xiang, Z. Facile synthesis of titania nanoparticles coated carbon nanotubes for selective enrichment of phosphopeptides for mass spectrometry analysis. Talanta 2013, 107, 30–35.
[6] Zian, L.; Jiang, Z.; Guo, L.; Zhi, T.; Xue, Y.; Zong, C. Negative Ion Laser Desorption/Ionization Time-of-Flight Mass Spectrometric Analysis of Small Molecules Using Graphitic Carbon Nitride Nanosheet Matrix. Analytical Chemistry 2015, 87, 8005-8012.
2. In Figure 2, the XRD patterns for all structures with Fe3O4are unclear. The signals are too weak. These curves must be improved.
Response: Thanks for your suggestion. The XRD pattern is revised in the manuscript and in the reply letter (Figure. 2a).
3. In scheme 1, the graphics are very difficult to follow. For example, why melamine is a lump; intermediate is rod-shaped; FeSO4has five bonds? What is the molecule consisted of 3 purple balls and 6 grey balls? This synthetic scheme must be coherent to the synthetic procedures reported.
Response: Thanks for your suggestion. The lump structure of melamine was observed by SEM (Supporting Information, 3. MATERIAL CHARACTERIZATION).
The formation process of the rod-like structure of the g-C3N4 intermediate has been mentioned in the manuscript (2. Results and discussions, 2.1 Characterization of the g-C3N4@Fe3O4 composite). The g-C3N4 intermediate formed through the hydrothermal reaction of melamine and urea. Among them, melamine and urea further formed nanopillar clusters through van der waals stacking reaction. With the occurrence of the hydrothermal reaction, the melamine molecule and the urea molecule underwent a self-assembly reaction, and the melamine is promoted to transform into a nano-columnar-like structure with a diameter of about 1.3 μm. In addition, the interior of the nanocolumnars was continuously stacked by hydrogen bonds, which maked the intermediates gradually pile up to form the aligned nanopillar-like cluster structure.
In addition, the revised synthetic road map is presented in the manuscript as well as in the reply (Scheme. 1). Among them, the cyan, green and purole balls represent FeCl3·6H2O, FeSO4·7H2O and Fe3O4, respectively. In the urea structure, the dark blue, light blue, red and pink balls represent N, H, C and O atoms, respectively.
4. In scheme 2, what are C6H14OS and C26H50O10? The fabrication of the wearable structure must be stated and discussed. Figure S8 and Figure S9 (curves only) should be moved to the main text.
Response: Thanks for your suggestion. C6H14OS and C26H50O10 are 6-mercapto-1-hexanol (MCH) and Tween-20, respectively(Supporting Information, 1.1Materials and Methods). We also revised the device construction schematic (Fig. 7a) in the manuscript.
The specific device construction procedure has been added in the manuscript. The device construction process is based on a PI flexible substrate chip with Au as the working electrode, Ag/AgCl as the reference electrode and Pt as the counter electrode. It was further completed by spinning g-C3N4@Fe3O4 sensing material and coating MCH blocking solution to block unreacted active sites and Tween-20 active agent solution.
In addition, we have revised Figure. S8 and S9 (curves only) in the supporting information to Figure. 7b and 7c in the manuscript.
5. More references on iron oxide-gold hybrid structures should also be cited.
Response: Thanks for your suggestion. According your suggestion, we have included references in the manuscript about potentiostatic deposition of gold nanoparticles.
[46] Ali, B.; Afsaneh, D.; Mohammad, M.; Fatemeh, M.; Reza, Z. Electrochemical deposition of gold nanoparticles on reduced graphene oxide modified glassy carbon electrode for simultaneous determination of levodopa, uric acid and folic acid. Journal of Electroanalytical Chemistry 2015, 736, 22-29.
[47] Jin, W.; Bei, Y.; Hui, W.; Ping, Y.; Yu, D. Highly sensitive electrochemical determination of Sunset Yellow based on gold nanoparticles/graphene electrode. Analytica Chimica Acta 2015, 893, 41-48.
In addition, we are still supporting information 1.EXPERIMENTAL PROCEDURES (1.12 Preparation of Work Electrode for Sensing) references references [1]-[3] about potentiostatic deposition of gold nanoparticles.
[1] Manikandan, S.; Durairaj, S.; Boateng, E.; Sidhureddy, B.; Chen, A. Electrochemical Detection of Nitrite Based on Co3O4-Au Nanocomposites for Food Quality Control. Journal of The Electrochemical Society 2021, 168, 107505.
[2] Cui’e, Z.; Bei, Y.; Duan, B.; Jin, W.; Shu, L.; Ping, Y.; Cai, W.; Yukihide, S.; Yukou, D. Electrochemical synthesis of gold nanoparticles decorated flower-like graphene for high sensitivity detection of nitrite. Journal of Colloid and Interface Science 2017, 488, 135-141.
[3] Mousavi-Majd, A.; Ghasemi, S.; Hosseini, R. Zeolitic imidazolate framework derived porous ZnO/Co3O4 incorporated with gold nanoparticles as ternary nanohybrid for determination of hydrazine. Journal of Alloys and Compounds 2022, 896, 162922.
6. The English usage must be improved. There are many typos in the manuscript, for example:Scheme 2, caption: diagaram; sewat
Response: Thanks for your suggestion. We're sorry to bother you with our error. We have made improvements to the English grammar of the article.
